# A Non-Asymptotic Analysis for
# Stein Variational Gradient Descent

**Anna Korba**
Gatsby Computational Neuroscience Unit
University College London
a.korba@ucl.ac.uk

**Adil Salim**
Visual Computing Center
KAUST
adil.salim@kaust.edu.sa

**Michael Arbel**
Gatsby Computational Neuroscience Unit
University College London
michael.n.arbel@gmail.com

**Giulia Luise**
Computer Science Department
University College London
g.luise16@ucl.ac.uk

**Arthur Gretton**
Gatsby Computational Neuroscience Unit
University College London
arthur.gretton@gmail.com

## Abstract

We study the Stein Variational Gradient Descent (SVGD) algorithm, which optimises a set of particles to approximate a target probability distribution $\pi \propto e^{-V}$ on $\mathbb{R}^d$. In the population limit, SVGD performs gradient descent in the space of probability distributions on the KL divergence with respect to $\pi$, where the gradient is smoothed through a kernel integral operator. In this paper, we provide a novel finite time analysis for the SVGD algorithm. We provide a descent lemma establishing that the algorithm decreases the objective at each iteration, and rates of convergence for the averaged Stein Fisher divergence (also referred to as Kernel Stein Discrepancy). We also provide a convergence result of the finite particle system corresponding to the practical implementation of SVGD to its population version.

## 1 Introduction

The task of sampling from a target distribution is common in Bayesian inference, where the distribution of interest is the posterior distribution of the parameters. Unfortunately, the posterior distribution is generally difficult to compute due to the presence of an intractable integral. This sampling problem can be formulated from an optimization point of view (Wibisono, 2018). We assume that the target distribution $\pi$ admits a density proportional to $\exp(-V)$ with respect to Lebesgue measure over $\mathcal{X} = \mathbb{R}^d$, where $V : \mathcal{X} \to \mathbb{R}$ is referred to as the potential function. In this setting, the target distribution $\pi$ is the solution to the optimization problem defined on the set $\mathcal{P}_2(\mathcal{X})$ of probability measures $\mu$ such that $\int \|x\|^2 d\mu(x) < \infty$ by:

$$\min_{\mu \in \mathcal{P}_2(\mathcal{X})} \mathrm{KL}(\mu|\pi), \tag{1}$$

where KL denotes the Kullback-Leibler divergence, and assuming $\pi \in \mathcal{P}_2(\mathcal{X})$. Many existing methods for the sampling task can be related to this optimization problem. Variants of the Langevin Monte Carlo algorithm (Durmus and Moulines, 2016; Dalalyan and Karagulyan, 2019) can be seen

as time-discretized schemes of the gradient flow of the relative entropy. These methods generate a Markov chain whose law converges to $\pi$ under mild assumptions, but the rates of convergence deteriorate quickly in high dimensions (Durmus et al., 2018b). Variational inference methods instead restrict the search space of problem (1) to a family of parametric distributions (Zhang et al., 2018; Ranganath et al., 2014). These methods are much more tractable in the large scale setting, since they benefit from efficient optimization methods (parallelization, stochastic optimization); however they can only return an approximation of the target distribution.

Recently, the Stein Variational Gradient Descent (SVGD) algorithm (Liu and Wang, 2016) was introduced as a non-parametric alternative to variational inference methods. It uses a set of interacting particles to approximate the target distribution, and applies iteratively to these particles a form of gradient descent of the relative entropy, where the descent direction is restricted to belong to a unit ball in a Reproducing Kernel Hilbert space (RKHS) (Steinwart and Christmann, 2008). In particular, this algorithm can be seen as a discretization of the gradient flow of the relative entropy on the space of probability distributions, equipped with a distance that depends on the kernel (Liu, 2017; Duncan et al., 2019). The empirical performance of this algorithm and its variants have been largely demonstrated in various tasks in machine learning such as Bayesian inference (Liu and Wang, 2016; Feng et al., 2017; Liu and Zhu, 2018; Detommaso et al., 2018), learning deep probabilistic models (Wang and Liu, 2016; Pu et al., 2017), or reinforcement learning (Liu et al., 2017). In the limit of infinite particles, the algorithm is known to converge to the target distribution under appropriate growth assumptions on the potential (Lu et al., 2019). Nonetheless, its non-asymptotic analysis remains incomplete: in particular, to the best of our knowledge, quantitative rates of convergence have yet to be obtained. The present paper aims at answering this question. Our first contribution is to provide in the infinite-particle regime a descent lemma showing that SVGD decreases at each iteration for a sufficiently small but constant step-size, with an analysis different from Liu (2017). We view this problem as an optimization problem over $\mathcal{P}_2(\mathcal{X})$ equipped with the Wasserstein distance, and use this framework and optimization techniques to obtain our results. Our second contribution is to provide in the finite particle regime, a propagation of chaos bound that quantifies the deviation of the empirical distribution of the particles to its population version.

This paper is organized as follows. Section 2 introduces the background needed on optimal transport, while Section 3 presents the point of view adopted to study SVGD in the infinite number of particles regime and reviews related work. Section 4 studies the continuous time dynamics of SVGD. Our main result is presented in Section 5, where we provide a descent lemma and rates of convergence for the SVGD algorithm. We also provide a convergence result of the finite particle system to its population version Section 6. The complete proofs and toy experiments are deferred to the appendix.

## 2 Preliminaries on optimal transport

Let $\mathcal{X} = \mathbb{R}^d$. We denote by $C^l(\mathcal{X})$ the space of $l$ continuously differentiable functions on $\mathcal{X}$. If $\psi : \mathcal{X} \to \mathbb{R}^p$, $p \geq 0$, is differentiable, we denote by $J\psi : \mathcal{X} \to \mathbb{R}^{p \times d}$ the Jacobian matrix of $\psi$. If $p = 1$, the gradient of $\psi$ denoted $\nabla \psi$ is seen as a column vector. Moreover, if $\nabla \psi$ is differentiable, the Jacobian of $\nabla \psi$ is the Hessian of $\psi$ denoted $H_\psi$. If $p = d$, $div(\psi)$ denotes the divergence of $\psi$, i.e., the trace of the Jacobian. The Hilbert-Schmidt norm of a matrix is denoted $\| \cdot \|_{HS}$ and the operator norm denoted $\| \cdot \|_{op}$.

### 2.1 The Wasserstein space and the continuity equation

In this section, we recall some background from optimal transport. The reader may refer to Ambrosio et al. (2008) for more details.

Consider the set $\mathcal{P}_2(\mathcal{X})$ of probability measures $\mu$ on $\mathcal{X}$ with finite second order moment. For any $\mu \in \mathcal{P}_2(\mathcal{X})$, $L^2(\mu)$ is the space of functions $f : \mathcal{X} \to \mathcal{X}$ such that $\int \|f\|^2 d\mu < \infty$. If $\mu \in \mathcal{P}_2(\mathcal{X})$, we denote by $\| \cdot \|_{L^2(\mu)}$ and $\langle \cdot, \cdot \rangle_{L^2(\mu)}$ respectively the norm and the inner product of the Hilbert space $L^2(\mu)$. Given a measurable map $T : \mathcal{X} \to \mathcal{X}$ and $\mu \in \mathcal{P}_2(\mathcal{X})$, we denote by $T_\# \mu$ the pushforward measure of $\mu$ by $T$, characterized by the transfer lemma $\int \phi(T(x))d\mu(x) = \int \phi(y)dT_\# \mu(y)$, for any measurable and bounded function $\phi$. Consider $\mu, \nu \in \mathcal{P}_2(\mathcal{X})$, the 2-nd order Wasserstein distance is defined by $W_2^2(\mu, \nu) = \inf_{s \in \mathcal{S}(\mu,\nu)} \int \|x - y\|^2 ds(x, y)$, where $\mathcal{S}(\mu, \nu)$ is the set of couplings between $\mu$ and $\nu$, i.e. the set of nonnegative measures $s$ over $\mathcal{X} \times \mathcal{X}$ such that $P_\# s = \mu$ (resp. $Q_\# s = \nu$) where $P : (x, y) \mapsto x$ (resp. $Q : (x, y) \mapsto y$) denotes the projection onto the first

(resp. the second) component. The Wasserstein distance is a distance over $\mathcal{P}_2(\mathcal{X})$. The metric space $(\mathcal{P}_2(\mathcal{X}), W_2)$ is called the Wasserstein space.

Let $T > 0$. Consider a weakly continuous map $\mu : (0, T) \to \mathcal{P}_2(\mathcal{X})$. The family $(\mu_t)_{t \in (0,T)}$ satisfies a continuity equation if there exists $(v_t)_{t \in (0,T)}$ such that $v_t \in L^2(\mu_t)$ and

$$\frac{\partial \mu_t}{\partial t} + div(\mu_t v_t) = 0 \tag{2}$$

holds in the distributional sense. A family $(\mu_t)_t$ satisfying a continuity equation with $\|v_t\|_{L^2(\mu_t)}$ integrable over $(0, T)$ is said absolutely continuous. Among the possible processes $(v_t)_t$, one has a minimal $L^2(\mu_t)$ norm and is called the velocity field of $(\mu_t)_t$. In a Riemannian interpretation of the Wasserstein space (Otto, 2001), this minimality condition can be characterized by $v_t$ belonging to the tangent space to $\mathcal{P}_2(\mathcal{X})$ at $\mu_t$ denoted $T_{\mu_t}\mathcal{P}_2(\mathcal{X})$, which is a subset of $L^2(\mu_t)$.

### 2.2 A functional defined over the Wasserstein space

Consider $\pi \propto \exp(-V)$ where $V : \mathcal{X} \to \mathbb{R}$ is a smooth function, i.e. $V$ is $C^2(\mathcal{X})$ and its Hessian $H_V$ is bounded from above. For any $\mu, \pi \in \mathcal{P}_2(\mathcal{X})$, the Kullback-Leibler divergence of $\mu$ w.r.t. $\pi$ is defined by

$$\mathrm{KL}(\mu|\pi) = \int \log\left(\frac{d\mu}{d\pi}(x)\right) d\mu(x)$$

if $\mu$ is absolutely continuous w.r.t. $\pi$ with Radon-Nikodym density $d\mu/d\pi$, and $\mathrm{KL}(\mu|\pi) = +\infty$ otherwise. Consider the functional $\mathrm{KL}(.|\pi) : \mathcal{P}_2(\mathcal{X}) \to [0, +\infty)$, $\mu \mapsto \mathrm{KL}(\mu|\pi)$ defined over the Wasserstein space. We shall perform differential calculus over this space for such a functional, which is a "powerful way of computing" (Villani, 2003, Section 8.2). If $\mu \in \mathcal{P}_2(\mathcal{X})$ satisfies some mild regularity conditions, the (Wasserstein) gradient of $\mathrm{KL}(.|\pi)$ at $\mu$ is denoted by $\nabla_{W_2} \mathrm{KL}(\mu|\pi) \in L^2(\mu)$ and defined by $\nabla \log\left(\frac{d\mu}{d\pi}\right)$. Moreover, the (Wasserstein) Hessian of $\mathrm{KL}(.|\mu)$ at $\mu$ is an operator over $T_\mu \mathcal{P}_2(\mathcal{X})$ defined by

$$\langle v, H_{\mathrm{KL}(.|\pi)}(\mu)v\rangle_{L^2(\mu)} = \mathbb{E}_{X \sim \mu}\left[\langle v(X), H_V(X)v(X)\rangle + \|Jv(X)\|_{HS}^2\right] \tag{3}$$

for any tangent vector $v \in T_\mu \mathcal{P}_2(\mathcal{X})$. Note that the Hessian of $\mathrm{KL}(.|\pi)$ is not bounded from above. An important property of the Wasserstein gradient is that it satisfies a chain rule. Let $(\mu_t)_t$ be an absolutely continuous curve s. t. $\mu_t$ has a density. Denote $(v_t)$ the velocity field of $(\mu_t)$. If $\varphi(t) = \mathrm{KL}(\mu_t|\pi)$, then under mild technical assumptions $\varphi'(t) = \langle v_t, \nabla_{W_2} \mathrm{KL}(\mu_t|\pi)\rangle_{L^2(\mu_t)}$ (see Ambrosio et al. (2008)).

## 3 Presentation of Stein Variational Gradient Descent (SVGD)

In this section, we present our point of view on SVGD in the infinite number of particles regime.

### 3.1 Kernel integral operator

Consider a positive semi-definite kernel $k : \mathcal{X} \times \mathcal{X} \to \mathbb{R}$ and $\mathcal{H}_0$ its corresponding RKHS of real-valued functions on $\mathcal{X}$. The space $\mathcal{H}_0$ is a Hilbert space with inner product $\langle \cdot, \cdot \rangle_{\mathcal{H}_0}$ and norm $\| \cdot \|_{\mathcal{H}_0}$ (see Smola and Scholkopf (1998)). Moreover, $k$ satisfies the reproducing property: $\forall f \in \mathcal{H}_0$, $f(x) = \langle f, k(x, .)\rangle_{\mathcal{H}_0}$. Denote by $\mathcal{H}$ the product RKHS consisting of elements $f = (f_1, \ldots, f_d)$ with $f_i \in \mathcal{H}_0$, and with a standard inner product $\langle f, g\rangle_{\mathcal{H}} = \sum_{i=1}^d \langle f_i, g_i\rangle_{\mathcal{H}_0}$. Let $\mu \in \mathcal{P}_2(\mathcal{X})$; the integral operator associated to kernel $k$ and measure $\mu$ denoted by $S_\mu : L^2(\mu) \to \mathcal{H}$ is

$$S_\mu f = \int k(x, \cdot)f(x)d\mu(x). \tag{4}$$

We make the key assumption that $\int k(x, x)d\mu(x) < \infty$ for any $\mu \in \mathcal{P}_2(\mathcal{X})$; which implies that $\mathcal{H} \subset L^2(\mu)$. Consider functions $f, g \in L_2(\mu) \times \mathcal{H}$ and denote the inclusion $\iota : \mathcal{H} \to L^2(\mu)$, with $\iota^* = S_\mu$ its adjoint. Then following e.g. (Steinwart and Christmann, 2008, Chapter 4), we have

$$\langle f, \iota g\rangle_{L^2(\mu)} = \langle \iota^* f, g\rangle_{\mathcal{H}} = \langle S_\mu f, g\rangle_{\mathcal{H}}. \tag{5}$$

When the kernel is integrally strictly positive definite, then $\mathcal{H}$ is dense in $L^2(\mu)$ for any probability measure $\mu$ (Sriperumbudur et al., 2011). We also define $P_\mu : L^2(\mu) \to L^2(\mu)$ the operator $P_\mu = \iota S_\mu$; notice that it differs from $S_\mu$ only in its range.

## 3.2 Stein Variational Gradient Descent

We can now present the Stein Variational Gradient Descent (SVGD) algorithm (Liu and Wang, 2016). The goal of this algorithm is to provide samples from a target distribution $\pi \propto \exp(-V)$ with positive density w.r.t. Lebesgue measure and known up to a normalization constant. Several point of views on SVGD have been adopted in the literature. In this paper, we view SVGD as an optimization algorithm (Liu, 2017) to minimize the Kullback-Leibler (KL) divergence w.r.t. $\pi$, see Problem (1). Denote $\mathrm{KL}(.|\pi) : \mathcal{P}_2(\mathcal{X}) \to [0, +\infty)$ the functional $\mu \mapsto \mathrm{KL}(\mu|\pi)$. More precisely, in order to obtain samples from $\pi$, SVGD applies a gradient descent-like algorithm to the functional $\mathrm{KL}(.|\pi)$. The standard gradient descent algorithm in the Wasserstein space applied to $\mathrm{KL}(.|\pi)$, at each iteration $n \geq 0$, is

$$\mu_{n+1} = \left( I - \gamma \nabla \log \left( \frac{\mu_n}{\pi} \right) \right)_{\#} \mu_n, \tag{6}$$

where $\gamma > 0$ is a step size and $I$ the identity map. This corresponds to a forward Euler discretization of the gradient flow of $\mathrm{KL}(.|\pi)$ (Wibisono, 2018), and can be seen as a Riemannian gradient descent where the exponential map at $\mu$ is the map $\phi \mapsto (I + \phi)_{\#}\mu$ defined on $L^2(\mu)$. Therefore, the gradient descent algorithm would require to estimate the density of $\mu_n$ based on samples, which can be demanding (though see Remark 1 below). We next examine the analogous SVGD iteration,

$$\mu_{n+1} = \left( I - \gamma P_{\mu_n} \nabla \log \left( \frac{\mu_n}{\pi} \right) \right)_{\#} \mu_n. \tag{7}$$

Instead of using $\nabla_{W_2} \mathrm{KL}(\mu_n|\pi)$ as the gradient, SVGD uses $P_{\mu_n} \nabla_{W_2} \mathrm{KL}(\mu_n|\pi)$. This can be seen as the gradient of $\mathrm{KL}(.|\pi)$ under the inner product of $\mathcal{H}$, since $\langle S_\mu \nabla_{W_2} \mathrm{KL}(\mu|\pi), v \rangle_{\mathcal{H}} = \langle \nabla_{W_2} \mathrm{KL}(\mu|\pi), \iota v \rangle_{L^2(\mu)}$ for any $v \in \mathcal{H}$. The important fact is that given samples of $\mu$, the evaluation of $P_\mu \nabla_{W_2} \mathrm{KL}(\mu|\pi)$ is simple. Indeed if $\lim_{\|x\| \to \infty} k(x, .)\pi(x) = 0$,

$$P_\mu \nabla \log \left( \frac{\mu}{\pi} \right)(\cdot) = - \int [\nabla \log \pi(x) k(x, \cdot) + \nabla_x k(x, \cdot)] d\mu(x), \tag{8}$$

using an integration by parts (see Liu (2017)).

**Remark 1.** An alternative sampling algorithm which does not imply to compute the exact gradient of the KL is the Unadjusted Langevin Algorithm (ULA). It is an implementable algorithm that computes a gradient step with $\nabla \log \pi$, and a flow step adding a Gaussian noise to the particles. However, it is not a gradient descent discretization; it rather corresponds to performing a Forward-Flow (FFl) discretization, which is biased (Wibisono, 2018, Section 2.2.2).

## 3.3 Stein Fisher information

The squared RKHS norm of the gradient $S_\mu \nabla \log(\frac{\mu}{\pi})$ is defined as the Stein Fisher Information:

**Definition 1.** Let $\mu \in \mathcal{P}_2(\mathcal{X})$. The *Stein Fisher Information* of $\mu$ relative to $\pi$ Duncan et al. (2019) is defined by :

$$I_{Stein}(\mu|\pi) = \| S_\mu \nabla \log \left( \frac{\mu}{\pi} \right) \|_{\mathcal{H}}^2. \tag{9}$$

**Remark 2.** Notice that since $P_\mu = \iota S_\mu$ with $\iota^* = S_\mu$, we can write $I_{Stein}(\mu|\pi) = \langle \nabla \log(\frac{\mu}{\pi}), P_\mu \nabla \log(\frac{\mu}{\pi}) \rangle_{L^2(\mu)}$.

In the literature the quantity (9) is also referred to as the squared Kernel Stein Discrepancy (KSD), used in nonparametric statistical tests for goodness-of-fit (Liu et al., 2016; Chwialkowski et al., 2016; Gorham and Mackey, 2017). The KSD provides a discrepancy between probability distributions, which depends on $\pi$ only through the score function $\nabla \log \pi$ that can be calculated without knowing the normalization constant of $\pi$. Whether the convergence of the KSD to zero, i.e. $I_{stein}(\mu_n|\pi) \to 0$ when $n \to \infty$ implies the weak convergence of $(\mu_n)$ to $\pi$ (denoted $\mu_n \to \pi$) depends on the choice of the kernel relatively to the target. This question has been treated in Gorham and Mackey (2017). Sufficient conditions include $\pi$ being distantly dissipative [1] which is similar to strong log concavity outside a bounded domain, and the kernel having a slow decay rate (e.g. being translation invariant with a non-vanishing Fourier transform, or $k$ being the inverse multi-quadratic kernel defined by

$k(x, y) = (c^2 + \|x - y\|_2^2)^\beta$ for $c > 0$ and $\beta \in [-1, 0]$). In these cases, $I_{stein}(\mu_n | \pi) \to 0$ implies $\mu_n \to \pi$.

In order to study the continuous time dynamics of SVGD, Duncan et al. (2019) introduced a kernel version of a log-Sobolev inequality (which usually upper bounds the KL by the Fisher divergence (Vempala and Wibisono, 2019)).

**Definition 2.** We say that $\pi$ satisfies the *Stein log-Sobolev inequality* with constant $\lambda > 0$ if:

$$\mathrm{KL}(\mu|\pi) \le \frac{1}{2\lambda} I_{Stein}(\mu|\pi). \tag{10}$$

The functional inequality (10) is not as well known and understood as the classical log-Sobolev inequality.[2] Duncan et al. (2019) provided a first investigation into when this condition might hold. They show that it *fails* to hold if the kernel is too regular w.r.t. $\pi$, more precisely for $k \in C^{1,1}(\mathcal{X} \times \mathcal{X})$, and if $\sum_{i=1}^d [(\partial_i V(x))^2 k(x, x) - \partial_i V(x)(\partial_i^1 k(x, x) + \partial_i^2 k(x, x)) + \partial_i^1 \partial_i^2 k(x, x)] d\pi(x) < \infty$, where $\partial_i^1$ and $\partial_i^2$ denote derivatives with respect to the first and second argument of $k$ respectively (Duncan et al., 2019, Lemma 36). This holds for instance in the case where $\pi$ has exponential tails and the derivatives of $k$ and $V$ grow at most at a polynomial rate. However, they provide interesting cases in dimension 1 where (10) holds, depending on $k$ and $\pi$. For instance, by choosing a nondifferentiable kernel that is adapted to the tails of the target $k(x, y) = \pi(x)^{-1/2} e^{-|x-y|} \pi(y)^{-1/2}$, and if $V''(x) + (V'(x))^2/2 \ge \tilde{\lambda} > 0$ for any $x \in \mathbb{R}$, then (10) holds with $\lambda = \min(1, \tilde{\lambda})$ (Duncan et al., 2019, Example 40). Conditions where (10) holds in higher dimensions are more challenging to establish, and are a topic of current research.

### 3.4 Related work

SVGD was originally introduced by Liu and Wang (2016), and was shown empirically to be competitive with state-of-the-art methods in Bayesian inference. Liu (2017) developed the first theoretical analysis and studied the weak convergence properties of SVGD. They showed that for any iteration, the empirical distribution of the SVGD samples (i.e., for a finite number of particles) weakly converges to the population distribution when the number of particles goes to infinity. In the infinite particle regime, they provided a descent lemma showing that the KL objective decreases at each iteration (see Remark 4). Finally, they derived the non-linear partial differential equation (PDE) that governs continuous time dynamics of SVGD, and provided a geometric intuition that interprets SVGD as a gradient flow of the KL divergence under a new Riemannian metric structure (the *Stein geometry*) induced by the kernel. Liu and Wang (2018) studied the fixed point properties of the algorithm for a finite number of particles, and showed that it exactly estimates expectations under the target distribution, for a set of functions called the Stein matching set, that are determined by the Stein operator (depending on the target distribution) and the kernel. In particular, they showed that by choosing linear kernels, SVGD can exactly estimate the mean and variance of Gaussian distributions when the number of particles is greater than the dimension. They further derived high probability bounds that bound the Kernel Stein Discrepancy between the empirical distribution and the target measure when the kernel is approximated with random features. Lu et al. (2019) studied the continuous time dynamics of SVGD in the infinite number of particles regime. They showed that the PDE governing continuous-time, infinite sample SVGD dynamics is well-posed, and that the law of the particle system (for a finite number of particles) is a weak solution of the equation, under appropriate growth conditions on the score function $\nabla \log \pi$, and they studied the regularity of the PDE. Finally, Duncan et al. (2019) investigated the contraction and equilibration properties of this PDE. In particular, they proposed conditions that induce exponential convergence to the equilibrium in continuous time, notably as the Stein log-Sobolev inequality, which relates the convexity of the KL objective to the *Stein geometry* (see Section 4). By contrast with Lu et al. (2019); Duncan et al. (2019), we develop a theoretical understanding of SVGD in *discrete time*, where to our knowledge rates of convergence have yet to be established.

## 4 Continuous-time dynamics of SVGD

This section defines and describes the SVGD dynamics in continuous time. Some of the results are already stated in Liu (2017) and Duncan et al. (2019) but are necessary to understand the discrete

time analysis. We provide intuitive sketches of the proof ideas in the main document, which exploit the differential calculus over the Wasserstein space. Detailed proofs are given in the Appendix.

The SVGD gradient flow is defined as the flow induced by the continuity equation (Liu, 2017):

$$\frac{\partial \mu_t}{\partial t} + div(\mu_t v_t) = 0, \qquad v_t := -P_{\mu_t} \nabla \log \left( \frac{\mu_t}{\pi} \right). \tag{11}$$

Equation (11) was shown to admit a unique and well defined solution (given an initial condition $\mu_0 \in \mathcal{P}_2(\mathcal{X})$) provided that some smoothness and growth assumptions on both kernel and target density $\pi$ are satisfied (Lu et al., 2019). Notice that the SVGD update (7) is a forward Euler discretization of (11). We propose to study the dissipation of the KL along the trajectory of the SVGD gradient flow. The Stein Fisher Information turns out to be the quantity that quantifies this dissipation, as stated in the next proposition.

**Proposition 1.** The dissipation of the KL along the SVGD gradient flow (11) is:

$$\frac{d \, \mathrm{KL}(\mu_t|\pi)}{dt} = -I_{Stein}(\mu_t|\pi). \tag{12}$$

*Proof.* Recall that $\nabla_{W_2} \mathrm{KL}(\mu|\pi) = \nabla \log(\frac{\mu}{\pi})$; using differential calculus in the Wasserstein space and the chain rule we have,

$$\frac{d \, \mathrm{KL}(\mu_t|\pi)}{dt} = \left\langle v_t, \nabla \log \left( \frac{\mu_t}{\pi} \right) \right\rangle_{L^2(\mu_t)} = - \left\| S_{\mu_t} \nabla \log \left( \frac{\mu_t}{\pi} \right) \right\|_{\mathcal{H}}^2. \qquad \square$$

Since $I_{Stein}(\mu|\pi)$ is nonnegative, Proposition 1 shows that the KL divergence with respect to $\pi$ decreases along the SVGD dynamics, i.e. the KL is a Lyapunov functional for the PDE (11). It can actually be proven that $I_{Stein}(\mu_t|\pi) \to 0$, as stated in the following proposition. Its proof is deferred to Section 11.1.

**Proposition 2.** Let $\mu_t$ be a solution of (11). Assume Assumption ($\mathbf{A}_1$), ($\mathbf{A}_2$), hold and that $\exists C > 0$ such that $\int \|x\| d\mu_t(x) < C$ for all $t \geq 0$. Then $I_{Stein}(\mu_t|\pi) \to 0$.

**Remark 3.** In the proof of Lu et al. (2019, Theorem 2.8), the authors show that $\mu_t$ converges weakly towards $\pi$ when $V$ grows at most polynomially. However, they implictly assumed that $I_{Stein}(\mu_t|\pi) \to 0$ which does not need to be true in general (Lesigne, 2010). It can actually be proven that $I_{Stein}(\mu_t|\pi) \to 0$ by controlling the oscillation of the $I_{Stein}(\mu_t|\pi)$ in time, using a semi-convexity result on the KL.

A second consequence of Proposition 1 is the following continuous time convergence rate for the average of $I_{Stein}(\mu_t|\pi)$. It is obtained immediately by integrating (12) and using the positivity of the KL.

**Proposition 3.** For any $t \geq 0$,

$$\min_{0 \leq s \leq t} I_{Stein}(\mu_s|\pi) \leq \frac{1}{t} \int_0^t I_{Stein}(\mu_s|\pi) ds \leq \frac{\mathrm{KL}(\mu_0|\pi)}{t}. \tag{13}$$

The convergence of $I_{Stein}(\mu_t|\pi)$ itself can be arbitrarily slow, however. To guarantee faster convergence rates of the SVGD dynamics, further properties are needed, such as convexity properties of the KL-divergence with respect to the Stein geometry. This is the purpose of the inequality (10) which implies exponential convergence of the SVGD gradient flow near equilibrium. Indeed, if $\pi$ satisfies the Stein log-Sobolev inequality, the Kullback-Leibler divergence converges exponentially fast along the SVGD dynamics.

**Proposition 4.** Assume $\pi$ satisfies the Stein log-Sobolev inequality with $\lambda > 0$. Then

$$\mathrm{KL}(\mu_t|\pi) \leq e^{-2\lambda t} \, \mathrm{KL}(\mu_0|\pi).$$

*Proof.* Combining (12) and (10) yields $\frac{d \, \mathrm{KL}(\mu_t|\pi)}{dt} \leq -2\lambda \, \mathrm{KL}(\mu_t|\pi)$. We conclude by applying Gronwall's lemma. $\qquad \square$

In the next section, we provide a non-asymptotic analysis for SVGD. Our first result holds without any convexity assumptions on the KL, but mainly under a smoothness assumption on $\pi$, while our second result leverages (10) to obtain rates of convergence.

# 5 Non-asymptotic analysis for SVGD

This section studies the SVGD dynamics in discrete time. Although one of the results echoes Liu (2017)[Theorem 3.3], we provide new convergence rates for the discrete time SVGD under mild conditions, and using different techniques: we return to this point in detail in Remark 4 below. Moreover, our proof technique is different. As in the previous section, we provide intuitive sketch of the proofs exploiting the differential calculus over the Wasserstein space. Each step of the proofs is rigourously justified in the Supplementary material.

Recall that the SVGD update is defined as (7). Let $\mu_0 \in \mathcal{P}_2(\mathcal{X})$ and assume that it admits a density. For every $n \geq 0$, $\mu_n$ is the distribution of $x_n$, where

$$x_{n+1} = x_n - \gamma P_{\mu_n} \nabla \log \left( \frac{\mu_n}{\pi} \right)(x_n), \quad x_0 \sim \mu_0. \tag{14}$$

This particle update leads to the finite particles implementation of SVGD, analysed in Section 6.

In this section, we analyze SVGD in discrete time, in the infinite number of particles regime (7). We propose to study the dissipation of the KL along the SVGD algorithm. The Stein Fisher Information once again quantifies this dissipation, as in the continuous time case. Before going further, note that discrete time analyses often require more assumptions that continuous time analyses. In optimization, these assumptions typically require some smoothness of the objective function. Here, we assume the following.

    ($\mathbf{A}_1$) Assume that $\exists B > 0$ s.t. for all $x \in \mathcal{X}$,
        $\|k(x,.)\|_{\mathcal{H}_0} \leq B$ and $\|\nabla_x k(x,.)\|_{\mathcal{H}} = (\sum_{i=1}^d \|\partial_{x_i} k(x_i,.)\|_{\mathcal{H}_0}^2)^{\frac{1}{2}} \leq B$.

    ($\mathbf{A}_2$) The Hessian $H_V$ of $V = -\log \pi$ is well-defined and $\exists M > 0$ s.t. $\|H_V\|_{op} \leq M$.

    ($\mathbf{A}_3$) Assume that $\exists$ is $C > 0$ s.t. $I_{Stein}(\mu_n|\pi) < C$ for all $n$.

Under Assumptions ($\mathbf{A}_1$) and ($\mathbf{A}_2$), a sufficient condition for Assumption ($\mathbf{A}_3$) is $\sup_n \int \|x\| \mu_n(x) dx < \infty$. Bounded moment assumptions such as these are commonly used in stochastic optimization, for instance in some analysis of the stochastic gradient descent (Moulines and Bach, 2011). Given our assumptions, we quantify the decreasing of the KL along the SVGD algorithm, also called a descent lemma in optimization.

**Proposition 5.** Assume that Assumptions ($\mathbf{A}_1$) to ($\mathbf{A}_3$) hold. Let $\alpha > 1$ and choose $\gamma \leq \frac{\alpha-1}{\alpha BC^{\frac{1}{2}}}$. Then:

$$\mathrm{KL}(\mu_{n+1}|\pi) - \mathrm{KL}(\mu_n|\pi) \leq -\gamma \left( 1 - \gamma \frac{(\alpha^2 + M)B^2}{2} \right) I_{stein}(\mu_n|\pi). \tag{15}$$

*Proof.* Our goal is to prove a discrete dissipation of the form $(\mathrm{KL}(\mu_{n+1}|\pi) - \mathrm{KL}(\mu_n|\pi))/\gamma \leq -I_{stein}(\mu_n|\pi) +$ error term. Our assumptions will control the error term. Fix $n \geq 0$ and denote $g = P_{\mu_n} \nabla \log(\frac{\mu_n}{\pi})$, $\phi_t = I - tg$ for $t \in [0, \gamma]$ and $\rho_t = (\phi_t)_{\#} \mu_n$. Note that $\rho_0 = \mu_n$ and $\rho_\gamma = \mu_{n+1}$.

Under our assumptions, one can show that for any $x \in \mathcal{X}$, $\|g(x)\|^2 \leq B^2 I_{Stein}(\mu_n|\pi)$ and $\|Jg(x)\|_{HS}^2 \leq B^2 I_{Stein}(\mu_n|\pi)$, using the reproducing property and Cauchy-Schwartz in $\mathcal{H}$. Hence, $\|tJg(x)\|_{op} < 1$ and $\phi_t$ is a diffeomorphism for every $t \in [0, \gamma]$. Moreover, $\|(J\phi_t)^{-1}(x)\|_{op} \leq \alpha$. Using (Villani, 2003, Theorem 5.34), the velocity field ruling the time evolution of $\rho_t$ is $w_t \in L^2(\rho_t)$ defined by $w_t(x) = -g(\phi_t^{-1}(x))$.

Denote $\varphi(t) = \mathrm{KL}(\rho_t|\pi)$. Using a Taylor expansion, $\varphi(\gamma) = \varphi(0) + \gamma \varphi'(0) + \int_0^\gamma (\gamma - t)\varphi''(t) dt$. We now identify each term. First,

$$\varphi(0) = \mathrm{KL}(\mu_n|\pi) \text{ and } \varphi(\gamma) = \mathrm{KL}(\mu_{n+1}|\pi).$$

Then, using the chain rule (Villani, 2003, Section 8.2),

$$\varphi'(t) = \langle \nabla_{W_2} \mathrm{KL}(\rho_t|\pi), w_t \rangle_{L^2(\rho_t)} \text{ and } \varphi''(t) = \langle w_t, Hess_{\mathrm{KL}(.|\pi)}(\rho_t) w_t \rangle_{L^2(\rho_t)}.$$

Therefore, $\varphi'(0) = -\langle \nabla \log \left( \frac{\mu_n}{\pi} \right), g \rangle_{L^2(\mu_n)} = -I_{Stein}(\mu_n|\pi)$. Moreover, $\varphi''(t) = \psi_1(t) + \psi_2(t)$, where

$$\psi_1(t) = \mathbb{E}_{x \sim \rho_t} \left[ \langle w_t(x), H_V(x) w_t(x) \rangle \right] \text{ and } \psi_2(t) = \mathbb{E}_{x \sim \rho_t} \left[ \|Jw_t(x)\|_{HS}^2 \right].$$

The first term $\psi_1(t)$ is bounded using Assumption $(\mathbf{A}_2)$, $\psi_1(t) \le M\|g\|_{L^2(\mu_n)}^2 \le MB^2 I_{Stein}(\mu_n|\pi)$. The second term $\psi_2(t)$ is the most challenging to bound as $\|Jw\|_{HS}$ cannot be controlled by $\|w\|$ for a general $w$. However, in our case, $w_t = -g \circ (\phi_t)^{-1}$, and $-Jw_t \circ \phi_t = Jg(J\phi_t)^{-1}$. Therefore, $\|Jw_t \circ \phi_t(x)\|_{HS}^2 \le \|Jg(x)\|_{HS}^2 \|(J\phi_t)^{-1}(x)\|_{op}^2 \le \alpha^2 B^2 I_{Stein}(\mu_n|\pi)$. Combining each of the quantity in the Taylor expansion gives the desired result. $\qquad\square$

Although the Hessian of $\mathrm{KL}(.|\pi)$ is not bounded over the whole tangent space, our proof relies on controlling the Hessian when restricted to $\mathcal{H}$. Since $I_{Stein}(\mu_n|\pi)$ is nonnegative, Proposition 5 shows that the KL divergence w.r.t. $\pi$ decreases along the SVGD algorithm, i.e. the KL is a Lyapunov functional for SVGD. A first consequence of Proposition 5 is the convergence of $I_{stein}(\mu_n|\pi)$ to zero, similarly to the continous time case, see Proposition 2. Indeed, the descent lemma implies that the sequence $I_{stein}(\mu_n|\pi)$ is summable and hence converges to zero. A second consequence of the descent lemma is the following discrete time convergence rate for the average of $I_{Stein}(\mu_n|\pi)$.

**Corollary 6.** Let $\alpha > 1$ and $\gamma \le \min\left(\frac{\alpha-1}{\alpha B C^{\frac{1}{2}}}, \frac{2}{(\alpha^2+M)B^2}\right)$ and $c_\gamma = \gamma\left(1 - \gamma\frac{(\alpha^2+M)B^2}{2}\right)$. Then,

$$\min_{k=1,\dots,n} I_{Stein}(\mu_n|\pi) \le \frac{1}{n}\sum_{k=1}^n I_{Stein}(\mu_k|\pi) \le \frac{\mathrm{KL}(\mu_0|\pi)}{c_\gamma n}.$$

We illustrate the validity of the rates of Corollary 6 with simple experiments provided Section 13. Corollary 6 provides a $\mathcal{O}(1/n)$ convergence rate for the arithmetic mean of the Kernel Stein Discrepancy (KSD) (which metricizes weak convergence in many cases, see section 3.3) between the iterates $\mu_n$ and $\pi$, under assumption $(\mathbf{A}_2)$ to assumption $(\mathbf{A}_3)$. It does not rely on Stein LSI nor on convexity of $V$, unlike most of the results on Langevin Monte Carlo (LMC) which assume either (standard) LSI or convexity of $V$ (Vempala and Wibisono, 2019; Durmus et al., 2019). To guarantee convergence rates of the SVGD algorithm in terms of the KL objective, further properties are needed. We discuss the difficulty of proving rates in KL in section 11.3.

**Remark 4.** A descent lemma was also obtained for SVGD in Liu (2017)[Theorem 3.3] under a boundedness condition of the KSD and the kernel. While we obtain similar conditions on the step size, our approach, shown in the proof sketch (and, in greater detail, the Appendix), gives clearer connections with Wasserstein gradient flows. More precisely, we prove Proposition 5 by performing differential calculus over the Wasserstein space. We are able to replace the boundedness condition on the KSD by a simple boundedness condition of the first moment of $\mu_n$ at each iteration, which echoes analyses of some optimization algorithms like Stochastic Gradient Descent (Moulines and Bach, 2011). Our construction also brings with it a simple yet informative perspective, arising from the optimization literature, into why SVGD actually satisfies a descent lemma. In optimization, it is well known that descent lemmas can be obtained under a boundedness condition on the Hessian matrix. Here, the Hessian operator of the KL at $\mu$ is an operator on $L^2(\mu)$; and yet, *this operator is not bounded* (Wibisono, 2018, Section 3.1.1). By restricting the Hessian operator to the RKHS however, and then using the reproducing property and our assumptions, the resulting Hessian operator is provably bounded under simple conditions on the kernel and $\pi$.

## 6  Finite number of particles regime

In this section, we investigate the deviation of the discrete distributions generated by the SVGD algorithm for a finite number of particles, to its population version. In practice, starting from $N$ i.i.d. samples $X_0^i \sim \mu_0$, SVGD algorithm updates the $N$ particles as follows :

$$X_{n+1}^i = X_n^i - \gamma P_{\hat{\mu}_n}\nabla \log\left(\frac{\hat{\mu}_n}{\pi}\right)(X_n^i), \qquad \hat{\mu}_n = \frac{1}{N}\sum_{j=1}^N \delta_{X_n^j}, \qquad (16)$$

where $\hat{\mu}_n$ denotes the empirical distribution of the interacting particles. Recall that $P_{\hat{\mu}_n}\nabla \log\left(\frac{\hat{\mu}_n}{\pi}\right)$ is well defined even if $\hat{\mu}_n$ is discrete.

In Liu (2017), the authors show that the empirical distribution of the SVGD samples weakly converge to its population limit for any iteration. More precisely, under the assumptions that $b(x,y) = \nabla \log \pi(x)k(x,y) + \nabla_1 k(x,y)$ is jointly Lipschitz and that $\hat{\mu}_0$ converges weakly to $\mu_0$ as $N \to \infty$

(which happens by drawing $N$ i.i.d. samples of $\mu_0$), for any $n \geq 0$, they show that $\hat{\mu}_n$ converges weakly to $\mu_n$. This happens as soon as Assumption $(\mathbf{A}_1),(\mathbf{A}_2),(\mathbf{B}_1),(\mathbf{B}_2)$ are satisfied (since the product of bounded Lipschitz functions is a Lipschitz function):

($\mathbf{B}_1$) Assume that $\exists C_V$ s.t. for all $x \in \mathcal{X}$, $\|V(x)\| \leq C_V$.

($\mathbf{B}_2$) Assume that $\exists D > 0$ s.t. $k$ is continuous on $\mathcal{X}$ and $D$-Lipschitz:
$|k(x, x') - k(y, y')| \leq D(\|x - y\| + \|x' - y'\|)$ for all $x, x', y, y' \in \mathcal{X}$,

and $k$ is continuously differentiable on $\mathcal{X}$ with $D$-Lipschitz gradient:
$\|\nabla k(x, x') - \nabla k(y, y')\| \leq D(\|x - y\| + \|x' - y'\|)$ for all $x, x', y, y' \in \mathcal{X}$.

Under these assumptions, we quantify the dependency on the number of particles in the following proposition.

**Proposition 7.** Let $n \geq 0$ and $T > 0$. Let $\mu_n$ and $\hat{\mu}_n$ be defined by (7) and (16) respectively. Under Assumption $(\mathbf{A}_1),(\mathbf{A}_2),(\mathbf{B}_1),(\mathbf{B}_2)$ for any $0 \leq n \leq \frac{T}{\gamma}$:

$$\mathbb{E}[W_2^2(\mu_n, \hat{\mu}_n)] \leq \frac{1}{2} \left( \frac{1}{\sqrt{N}} \sqrt{var(\mu_0)} e^{LT} \right) (e^{2LT} - 1)$$

where $L$ is a constant depending on $k$ and $\pi$.

Proposition 7, whose proof is provided section 11.4, uses techniques from Jourdain et al. (2007). It is a non-asymptotic result in the sense that it provides an explicit bound. However, it is not a bound that helps quantify the rate of minimization of the objective function, but a bound between the population distribution $\mu_n$ and its particle approximation $\hat{\mu}_n$. Such results are referred to as *propagation of chaos* in the PDE literature, where having the constant $C$ depending on $T$ is common. Getting a similar bound with $C$ not depending on $T$ would be a much stronger result referred to as *uniform in time propagation of chaos*. Such results, which are subject to active research in PDE, are hard to obtain. Among the recent exceptions is Durmus et al. (2018a) who consider the process $dx_t = -\nabla U(x_t) - \nabla W * \mu_t(x_t)dt$ and manage to prove such results when $U$ is strictly convex outside of a ball. However in SVGD (see (8)), the attractive force $\nabla \log \pi(x)k(x, .)$ cannot be written as the gradient of a confinement potential $U : \mathbb{R}^d \to \mathbb{R}$ in general. Hence these results do not apply, and the convergence rate for SVGD using $\hat{\mu}_n$ remains an open problem.

# 7  Conclusion

In this paper, we provide a non-asymptotic analysis for the SVGD algorithm. Our results build upon the connection of SVGD with gradient descent in the Wasserstein space (Liu, 2017). In establishing these results, we draw on perspectives and techniques used to establish convergence in optimization. Several questions remain open. Firstly, the question of deriving rates of convergence of SVGD (in the infinite particle regime) in terms of the Kullback-Leibler objective, when the potential $V$ is convex or when $\pi$ satisfies some log Sobolev inequality. Secondly, the question of deriving a unified bound for the convergence of $\hat{\mu}_n$ to $\pi$ (decreasing as the number of iterations $n$ and number of particles $N$ go to infinity). This would require to obtain a uniform in time propagation of chaos result for the SVGD particle system. Finally, another further direction would be to study SVGD dynamics when the kernel depends on the current distribution. These kind of dynamics arise in black-box variational inference and Generative Adversarial Networks (Chu et al., 2020) (in which case the kernel is the neural tangent kernel introduced by Jacot et al. (2018)).

# 8  Broader impact

This paper aims at bringing more theoretical understanding to the Stein Variational Gradient Descent algorithm. This algorithm is widely used by machine learning practitioners but its non asymptotic properties are not as well-known as the ones of the Langevin Monte Carlo algorithm which can be considered as its competitor.

# 9 Funding disclosure

AK, MA, and AG thank the Gatsby Charitable Foundation for the financial support.

## Footnotes

[1]i.e. such that $\liminf_{r \to \infty} \kappa(r) > 0$ for $\kappa(r) = \inf\{-2\langle \nabla \log \pi(x) - \nabla \log \pi(y), x - y \rangle / \|x - y\|_2^2; \|x - y\|_2 = r\}$. This includes finite Gaussian mixtures with common covariance and all distributions strongly log-concave outside of a compact set, including Bayesian linear, logistic, and Huber regression posteriors with Gaussian priors.

[2]i.e. $\mathrm{KL}(\mu|\pi) \le 1/2\lambda \|\nabla \log(\frac{\mu}{\pi})\|_{L^2(\mu)}^2$, which holds for instance as soon as $V$ is $\lambda$-strongly convex.

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
