[Supplementary Material]

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

## 10 Background

### 10.1 Dissipation of the KL

The time derivative, or the dissipation of the KL divergence along any flow is given by :

$$\frac{d}{dt}\,\mathrm{KL}(\mu_t|\pi) = \frac{d}{dt}\int \mu_t \log\left(\frac{\mu_t}{\pi}\right)dx = \int \frac{\partial \mu_t}{\partial t}\log\left(\frac{\mu_t}{\pi}\right)dx \tag{17}$$

since the second part of the chain rule is null :

$$\int \mu_t \frac{\partial}{\partial t}\log\left(\frac{\mu_t}{\pi}\right)dx = \int \frac{\partial \mu_t}{\partial t}dx = \frac{d}{dt}\int \mu_t dx = 0.$$

Moreover, if $\mu_t$ satisfies a continuity equation of the form :

$$\frac{\partial \mu_t}{\partial t} + div(\mu_t v_t) = 0$$

where $v_t$ is called the velocity field, then by an integration by parts :

$$\frac{d}{dt}\,\mathrm{KL}(\mu_t|\pi) = -\int div(\mu_t(x)v_t(x))\log\left(\frac{\mu_t}{\pi}\right)dx$$

$$= \int v_t(x)\nabla\log\left(\frac{\mu_t}{\pi}\right)(x)\mu_t(x)dx = \langle v_t, \nabla\log\left(\frac{\mu_t}{\pi}\right)\rangle_{L^2(\mu_t)}. \tag{18}$$

### 10.2 Descent lemma for Gradient Descent in $\mathbb{R}^d$

In this section we show how to obtain a descent lemma for the gradient descent algorithm. We do not claim any generality here, the goal of this section is to provide an intuition behind the proof of Proposition 5 for SVGD.

Consider $F : \mathbb{R}^d \to \mathbb{R}$ a $C^2(\mathbb{R}^d)$ function with Hessian $H_F$, and the gradient descent algorithm written at iteration $n+1$:

$$x_{n+1} = x_n - \gamma\nabla F(x_n). \tag{19}$$

Consider $n \geq 0$ fixed. For every $t \geq 0$, denote $x(t) = x_n - t\nabla F(x_n)$. Then, $x(0) = x_n$ and $x(\gamma) = x_{n+1}$. We assume that there exists $M \geq 0$ such that for every $t \geq 0$, $\|H_F(x(t))\| \leq M$.

Denote $\varphi(t) = F(x(t))$. Using Taylor expansion,

$$\varphi(\gamma) = \varphi(0) + \gamma\varphi'(0) + \int_0^\gamma (\gamma - t)\varphi''(t)dt. \tag{20}$$

Denote by $\dot{x}$ the derivative of $x$. We now identify each term. First, $\varphi(0) = F(x_n)$ and $\varphi(\gamma) = F(x_{n+1})$. Second, $\varphi'(0) = \langle \nabla F(x(0)), \dot{x}(0)\rangle = \langle \nabla F(x(0)), -\nabla F(x_n)\rangle = -\|\nabla F(x_n)\|^2$. Finally, since $\ddot{x} = 0$,

$$\varphi''(t) = \langle \dot{x}(t), H_F(x(t))\dot{x}(t)\rangle \leq M\|\dot{x}(t)\|^2 = M\|\nabla F(x_n)\|^2. \tag{21}$$

Therefore

$$F(x_{n+1}) \leq F(x_n) - \gamma\|\nabla F(x_n)\|^2 + M\int_0^\gamma (\gamma - t)\|\nabla F(x_n)\|^2 dt$$

$$\leq F(x_n) - \gamma\|\nabla F(x_n)\|^2 + \frac{M\gamma^2}{2}\|\nabla F(x_n)\|^2. \tag{22}$$

## 11 Proofs

### 11.1 Proof of Proposition 2

**Proposition 8.** Under Assumption $(\mathbf{A}_1)$, $(\mathbf{A}_2)$, and assuming $\exists C > 0$ such that $\int \|x\|d\mu_t(x) < C$ for all $t \geq 0$, there exists $\lambda \in \mathbb{R}^+$ such that:

$$\left|\frac{dI_{Stein}(\mu_t|\pi)}{dt}\right| \leq \lambda I_{Stein}(\mu_t|\pi). \tag{23}$$

*Proof.* We first need to compute $D_t = \frac{dI_{Stein}(\mu_t|\pi)}{dt}$. We denote by $v_t = S_{\mu_t}\nabla\log(\frac{\mu_t}{\pi})$. Recalling that $I_{Stein}(\mu_t|\pi) = \sum_{i=1}^d \|v_t^i\|_{\mathcal{H}_0}^2$ we have by differentiation that:

$$D_t = 2\sum_{i=1}^d \langle v_t^i, \frac{dv_t^i}{dt}\rangle_{\mathcal{H}_0} \tag{24}$$

We thus need to compute each component $\frac{dv_t^i}{dt}$. Those are given by direct calculation:

$$\frac{dv_t^i}{dt}(x) = \int [\partial_i \log \pi(x')k(x',x) + \partial_i k(x',x)]\frac{d\mu_t(x')}{dt}dx'$$

$$= -\int \langle \nabla[\partial_i \log \pi(x')k(x',x) + \partial_i k(x',x)], v_t(x')\rangle d\mu_t(x')$$

$$= -\int \sum_{i,j} \left[\partial_i\partial_j \log \pi(x')k(x',x) + \partial_i \log \pi(x')\partial_j k(x',x) + \partial_j\partial_i k(x',x)\right] v_t^j(x')d\mu_t(x').$$

where the second line uses an integration by parts. Hence by using the reproducing property,

$$D_t = 2\int \sum_{i,j} \left[\partial_i\partial_j \log \pi(x')v_t^i(x') + \partial_i \log \pi(x')\partial_j v_t^i(x') + \partial_j\partial_i v_t^i(x')\right] v_t^j(x')d\mu_t(x')$$

We will use the reproducing property recalling that each component $v_t^i$ is an element of the RKHS $\mathcal{H}_0$, i.e: $v_t^i(x) = \langle v_t^i, k(x,.)\rangle_{\mathcal{H}_0}$, hence:

$$D_t = 2\sum_{i,j}\langle v_t, A_{i,j}v_t\rangle_{\mathcal{H}_0}, \tag{25}$$

where $A_{i,j}$ are operators given by:

$$A_{i,j} = \int k(x',.) \otimes k(x,.)\partial_i\partial_j \log \pi(x')d\mu_t(x)d\mu_t(x')$$

$$+ \int \partial_i k(x'.) \otimes k(x,.)\partial_i \log \pi(x')d\mu_t(x)d\mu_t(x')$$

$$+ \int \partial_i k(x',.) \otimes \partial_j k(x,.)d\mu_t(x)d\mu_t(x').$$

We need to show that the $A_{i,j}$ have a bounded Hilbert-Schmidt norm at all times $t$. Indeed, if $\|A_{i,j}\|_{HS} \leq R$ for some $R > 0$, then we directly conclude that:

$$|D_t| \leq dR\sum_{i=1}^{d}\|v_t^i\|_{\mathcal{H}_0}^2 = dRI_{Stein}(\mu_t|\pi). \tag{26}$$

By assumptions on the kernel and Hessian of $\log \pi$ we have that:

$$\|A_{i,j}\|_{HS} \leq \int \|k(x',.)\|_{\mathcal{H}_0}|\partial_i\partial_j \log \pi(x')|d\mu_t(x')\int \|k(x,.)\|_{\mathcal{H}_0}d\mu_t(x)$$

$$+ \int \|\partial_i k(x',.)\|_{\mathcal{H}_0}|\partial_i \log \pi(x')|d\mu_t(x')\int \|k(x,.)\|_{\mathcal{H}_0}d\mu_t(x)$$

$$+ \left(\int \|\partial_i k(x',.)\|_{\mathcal{H}_0}d\mu_t(x')\right)^2$$

We recall that by assumption $\|k(x,.)\|_{\mathcal{H}_0} \leq B$, $\|\partial_i k(x',.)\|_{\mathcal{H}_0} \leq B$ and $\|H_{\log \pi}(x)\|_{op} \leq M$. Hence, we have:

$$\|A_{i,j}\|_{HS} \leq B^2(M + 1 + \int |\partial_i \log \pi(x)|d\mu_t(x)). \tag{27}$$

It remains to control $\partial_i \log \pi(x)$. This can be done under the additional assumption:

$$\int \|x\|d\mu_t(x) < C, \qquad \forall t \geq 0, \tag{28}$$

for some positive constant $C$. Hence, we have:

$$|\partial_i \log \pi(x)| \leq |\partial_i \log \pi(0)| + M\|x\|. \tag{29}$$

We finally get:

$$\|A_{i,j}\|_{HS} \leq B^2(M + 1 + MC + |\partial_i \log \pi(0)|) \tag{30}$$

Denoting $\lambda = dB^2(M + 1 + MC + |\partial_i \log \pi(0)|)$ gives the desired result.

$\square$

Recall, from the dissipation (Proposition 1) that $\mathrm{KL}(\mu_t|\pi) \leq \mathrm{KL}(\mu_0|\pi)$. Since $\rho \mapsto \mathrm{KL}(\rho|\pi)$ is weakly coercive (i.e., has compact sub-level sets in the weak topology, (van Erven and Harremoës, 2014, Theorem 20)), the family $(\mu_t)$ is weakly relatively compact. Besides, $I_{Stein}(\rho|\pi)$ is weakly continuous, therefore its supremum over the weakly relatively compact set $(\mu_t)$ is finite: $\sup_t I_{Stein}(\mu_t|\pi) < \infty$. Therefore, there exists $L \geq 0$ such that $|\frac{d}{dt} I_{Stein}(\mu_t|\pi)| \leq L$.

We can now show that $I_{Stein}(\mu_t|\pi)$ converges to 0. Indeed, otherwise we would have a sequence $t_k \to \infty$ such that $I_{Stein}(\mu_{t_k}|\pi) > \varepsilon > 0$. Moreover, since $I_{Stein}(\mu_t|\pi)$ has bounded time derivative, it is uniformly $L$-Lipschitz. There exists a sequence of intervals $I_k$ of length $\frac{\varepsilon}{L}$ centered at $t_k$ (that we can assume disjoints without loss of generality since $t_k \to \infty$), such that $I_{Stein}(\mu_t|\pi) \geq \frac{\varepsilon}{2}$ for every $t \in I_k$. Now, integrating the dissipation (see Proposition 1) over $\mathbb{R}^+$ we get:

$$\mathrm{KL}(\mu_0|\pi) - \mathrm{KL}(\mu_t|\pi) = \int_0^t I_{Stein}(\mu_s|\pi)ds \geq \sum_{k, t_k \leq t} \frac{\varepsilon^2}{2L}. \tag{31}$$

The above sum diverges as $t$ goes to infinity since $t_k \to +\infty$. This is in contradiction with $\mathrm{KL}(\mu_0|\pi) < \infty$. Hence, $I_{Stein}(\mu_t|\pi) \to 0$.

## 11.2 Proof of Proposition 5

We justify each step of the sketch of the proof of Section 10.2.

Consider $n \geq 0$ fixed and $\gamma \leq \frac{\alpha-1}{\alpha B C^{\frac{1}{2}}}$. Denote $g = P_{\mu_n} \nabla \log\left(\frac{\mu_n}{\pi}\right)$ and for every $t \in [0, \gamma]$, $\phi_t = (I - tg)$. Denote $\rho_t = \phi_{t\#}\mu_n$. Then, $\rho_0 = \mu_n$ and $\rho_\gamma = \mu_{n+1}$.

**Lemma 9.** Suppose Assumption ($\mathbf{A}_1$) holds, i.e. the kernel and its gradient are bounded by some positive constant $B$. Then for any $x \in \mathcal{X}$:

$$\|g(x)\| \leq B I_{Stein}(\mu_n|\pi)^{\frac{1}{2}} \tag{32}$$

$$\|Jg(x)\|_{HS} \leq B I_{Stein}(\mu_n|\pi)^{\frac{1}{2}} \tag{33}$$

*Proof.* This is a consequence of the reproducing property and Cauchy-Schwarz inequality in the RKHS space. Let $g' = S_{\mu_n} \nabla \log(\frac{\mu_n}{\pi})$, hence for any $x \in \mathcal{X}$, $g(x) = g'(x)$ and:

$$\|g(x)\|^2 = \sum_{i=1}^d \langle k(x,.), g'_i\rangle_{\mathcal{H}_0}^2 \leq \|k(x,.)\|_{\mathcal{H}_0}^2 \|g'\|_{\mathcal{H}}^2 \leq B^2 I_{Stein}(\mu_n|\pi).$$

Similarly:

$$\|Jg(x)\|_{HS}^2 = \sum_{i,j=1}^d \left|\frac{\partial g_i(x)}{\partial x_j}\right|^2 = \sum_{i,j=1}^d \langle \partial_{x_j} k(x,.), g'_i\rangle_{\mathcal{H}_0} \leq \sum_{i,j=1}^d \|\partial_{x_j} k(x,.)\|_{\mathcal{H}_0}^2 \|g'_i\|_{\mathcal{H}_0}^2$$
$$= \|\nabla k(x,.)\|_{\mathcal{H}}^2 \|g'\|_{\mathcal{H}}^2 \leq B^2 I_{Stein}(\mu_n|\pi).$$

$\square$

**Lemma 10.** Suppose that Assumption ($\mathbf{A}_1$) and Assumption ($\mathbf{A}_3$) hold. Then, for any $x \in \mathcal{X}$, $\|tJg(x)\|_{op} \leq tB\sqrt{C}$ and for every $t < \frac{1}{B\sqrt{C}}$, $\phi_t$ is a diffeomorphism. Moreover, $\|(J\phi_t(x))^{-1}\|_{op} \leq \alpha$.

*Proof.* First, by Lemma 9 and Assumption ($\mathbf{A}_3$) we have $\|Jg(x)\|_{op} \leq \|Jg(x)\|_{HS} \leq B\sqrt{C}$. If $t < \frac{1}{B\sqrt{C}}$, then $\|tJg(x)\|_{op} < 1$. Therefore, $J(\phi_t)(x) = I - tJg(x)$ is regular for every $x$ and $\phi_t$ is a diffeomorphism. Moreover,

$$\|(J\phi_t(x))^{-1}\|_{op} \leq \sum_{k=0}^\infty \|tJg(x)\|_{op}^k \leq \sum_{k=0}^\infty \|tJg(x)\|_{HS}^k \leq \sum_{k=0}^\infty (tB\sqrt{C})^k \leq \alpha, \tag{34}$$

where we used $\gamma \leq \frac{\alpha-1}{\alpha B C^{\frac{1}{2}}}$. $\square$

Denote $\varphi(t) = \mathrm{KL}(\rho_t|\pi)$. Using Taylor expansion,

$$\varphi(\gamma) = \varphi(0) + \gamma\varphi'(0) + \int_0^\gamma (\gamma - t)\varphi''(t)dt. \tag{35}$$

We now identify each term. First, $\varphi(0) = \mathrm{KL}(\mu_n|\pi)$ and $\varphi(\gamma) = \mathrm{KL}(\mu_{n+1}|\pi)$.

To compute $\varphi'(t)$ and $\varphi''(t)$ we have two options. Either we check the assumptions of the optimal transport theorems allowing to apply the chain rule Villani (2003); Ambrosio et al. (2008), or we do a direct computation. The latter is preferred, although differential calculus over the Wasserstein space is a powerful way to guess the formulas.

**Lemma 11.** Denote $w_t(x) = -g(\phi_t^{-1}(x))$. Then,

$$\varphi'(0) = \langle \nabla_{W_2} \mathrm{KL}(\rho_0|\pi), w_0 \rangle_{L^2(\mu_n)} = -I_{Stein}(\mu_n|\pi),$$

and,

$$\varphi''(t) = \langle w_t, Hess_{\mathrm{KL}(.|\pi)}(\rho_t) w_t \rangle_{L^2(\rho_t)} = \int \left[ \|Jg(x)(J\phi_t(x))^{-1}\|_{HS}^2 + \langle g(x), H_V(\phi_t(x))g(x)\rangle \right] \mu_n(x)dx.$$

*Proof.* We know by Lemma 10 that $\phi_t$ is a diffeomorphism, therefore, $\rho_t$ admits a density given by the change of variables formula:

$$\rho_t(x) = |J\phi_t(\phi_t^{-1}(x))|^{-1}\mu_n(\phi_t^{-1}(x)). \tag{36}$$

Using the transfer lemma with $\rho_t = \phi_{t\#}\mu_n$, $\varphi(t)$ is given by:

$$\varphi(t) = \int \log\left(\frac{\rho_t(y)}{\pi(y)}\right) \rho_t(y)dy$$

$$= \int \log\left(\frac{\mu_n(x)|J\phi_t(x)|^{-1}}{\pi(\phi_t(x))}\right) \mu_n(x)dx.$$

We can now take the time derivative of $\varphi(t)$ which gives:

$$\varphi'(t) = -\int tr\left(J\phi_t(x)^{-1}\frac{dJ\phi_t(x)}{dt}\right)\mu_n(x)dx - \int \langle \nabla \log \pi(\phi_t(x)), \frac{d\phi_t(x)}{dt}\rangle \mu_n(x)dx.$$

Hence, we can use the explicit expression of $\phi_t$ to write:

$$\varphi'(t) = \int tr(J\phi_t(x)^{-1}Jg(x))\mu_n(x)dx + \int \langle \nabla \log \pi(\phi_t(x)), g(x)\rangle \mu_n(x)dx.$$

The Jacobian at time $t = 0$ is simply equal to the identity since $\phi_0 = I$. It follows that $tr(J\phi_0(x)^{-1}Jg(x)) = tr(Jg(x)) = div(g)(x)$ by definition of the divergence operator. Using an integration by parts:

$$\varphi'(0) = -\int \left[-div(g)(x) - \langle \nabla \log \pi(x), g(x)\rangle\right]\mu_n(x)dx$$

$$= -\int \langle \nabla \log\left(\frac{\mu_n}{\pi}\right)(x), g(x)\rangle \mu_n(x)dx = -I_{Stein}(\mu_n|\pi).$$

Now, we prove the second statement. First,

$$\varphi''(t) = \int \left[tr((Jg(x)(J\phi_t(x))^{-1})^2) + \langle g(x), H_V(\phi_t(x))g(x)\rangle\right]\mu_n(x)dx.$$

Since $Jg(x)$ and $J\phi_t(x)$ commutes, $tr((Jg(x)(J\phi_t(x))^{-1})^2) = \|Jg(x)(J\phi_t(x))^{-1}\|_{HS}^2$. Moreover, using the chain rule,

$$-Jw_t(x) = J(g \circ \phi_t^{-1})(x) = Jg(\phi_t^{-1}(x))J(\phi_t^{-1})(x) = Jg(\phi_t^{-1}(x))(J\phi_t)^{-1}(\phi_t^{-1}(x)). \tag{37}$$

Therefore, $\|Jg(x)(J\phi_t(x))^{-1}\|_{HS}^2 = \|Jw_t(\phi_t(x))\|_{HS}^2$, which proves the second part of the second statement. Using the transfer lemma,

$$\varphi''(t) = \int \left[\|Jw_t(y)\|_{HS}^2 + \langle w_t(y), H_V(y)w_t(y)\rangle\right]\rho_t(y)dy$$

$$= \langle w_t, Hess_{\mathrm{KL}(.|\pi)}(\rho_t)w_t\rangle_{L^2(\rho_t)},$$

which concludes the proof. $\qquad\square$

Denote

$$\psi_1(t) = \int \left[\|Jg(x)(J\phi_t(x))^{-1}\|_{HS}^2\right]\mu_n(x)dx \quad \text{and} \quad \psi_2(t) = \int \langle g(x), H_V(\phi_t(x))g(x)\rangle \mu_n(x)dx.$$

Then, $\varphi''(t) = \psi_1(t) + \psi_2(t)$. We bound $\psi_1$ and $\psi_2$ separately. First, since the potential $V$ is $M$-smooth,

$$\psi_2(t) \leq M \int \|g(x)\|^2 \mu_n(x)dx \leq MB^2 I_{Stein}(\mu_n|\pi),$$

by using Lemma 9. Now, we bound $\psi_1(t)$ using Lemma 10 and 9:

$$\|Jg(x)(J\phi_t(x))^{-1}\|_{HS}^2 \leq \|Jg(x)\|_{HS}^2\|(J\phi_t(x))^{-1}\|_{op}^2 \leq \alpha^2 B^2 I_{Stein}(\mu_n|\pi). \tag{38}$$

Finally, $\varphi''(t) \leq (\alpha^2 + M)B^2 I_{Stein}(\mu_n|\pi)$. Plugging into (35) gives the result.

## 11.3 About combining the Stein log Sobolev assumption and a descent lemma

An insight deriving from the optimization perspective is that linear rates could be obtained by combining a descent result such as in proposition 5 and a Polyak-Lojasiewicz condition on the objective function Karimi et al. (2016). In our case, the latter condition corresponds to the Stein log Sobolev inequality from Duncan et al. (2019). Using the descent Proposition 5 and the Stein log Sobolev inequality (10) we would have that:

$$\text{KL}(\mu_{n+1}|\pi) - \text{KL}(\mu_n|\pi) \leq -c_\gamma I_{Stein}(\mu_n|\pi) \leq -2c_\gamma \lambda \, \text{KL}(\mu_n|\pi),$$

hence $\text{KL}(\mu_{n+1}|\pi) \leq (1-2c_\gamma\lambda)\,\text{KL}(\mu_n|\pi)$ which would result by iteration in a linear rate for the KL objective. *However*, it seems impossible to combine the assumptions needed for our descent lemma, in particular about the kernel and its derivative being bounded, while being able to asssume that the Stein log Sobolev inequality holds. It seems that no such $\pi$ and $k$ exist (at least for $\mathcal{X} = \mathbb{R}^d$). Given that both the kernel and its derivative are bounded, equation

$$\int \sum_{i=1}^d [(\partial_i V(x))^2 k(x,x) - \partial_i V(x)(\partial_i^1 k(x,x) + \partial_i^2 k(x,x)) + \partial_i^1 \partial_i^2 k(x,x)]d\pi(x) < \infty$$

reduces to a property on $V$ which, as far as we can tell, always holds; and this implies that Stein LSI does not hold (see (Duncan et al., 2019, Lemma 36)). For instance, even when $V = -\log(\text{cauchy})$ or $V = -\log(\text{student})$ the negative log densities of a Cauchy or Student distribution, we quickly find that the resulting expectations are bounded hence Stein LSI does not hold.

## 11.4 Proof of Proposition 7

Introduce the system of $N$ *independent* particles:

$$\bar{X}_{n+1}^i = \bar{X}_n^i - \gamma P_{\mu_n} \nabla \log\left(\frac{\mu_n}{\pi}\right)(\bar{X}_n^i), \quad \bar{X}_0^i \sim \mu_0. \tag{39}$$

By definition, $(\bar{X}_n^i)_{i=1}^N$ are i.i.d. samples from $\mu_n$. Let $c_n = \left(\frac{1}{N}\sum_{i=1}^N \mathbb{E}[\|\bar{X}_n^i - X_n^i\|^2]\right)^{\frac{1}{2}}$. Notice that $c_n \geq W_2(\mu_n, \hat{\mu}_n)$ since the 2-Wasserstein is the infimum over the couplings between $\mu_n$ and $\bar{\mu}_n$. At time $n+1$, we have:

$$
\begin{aligned}
c_{n+1} &= \frac{1}{\sqrt{N}}\left(\sum_{i=1}^N \mathbb{E}[\|X_{n+1}^i - \bar{X}_{n+1}^i\|^2]\right)^{\frac{1}{2}} \\
&= \frac{1}{\sqrt{N}}\left(\sum_{i=1}^N \mathbb{E}[\|X_n^i - \bar{X}_n^i - \gamma(P_{\hat{\mu}_n}\nabla\log(\frac{\hat{\mu}_n}{\pi})(X_n^i) - P_{\mu_n}\nabla\log(\frac{\mu_n}{\pi})(\bar{X}_n^i))\|^2]\right)^{\frac{1}{2}} \\
&\leq c_n + \frac{\gamma}{\sqrt{N}}\left(\sum_{i=1}^N \mathbb{E}[\|P_{\hat{\mu}_n}\nabla\log(\frac{\hat{\mu}_n}{\pi})(X_n^i) - P_{\mu_n}\nabla\log(\frac{\mu_n}{\pi})(\bar{X}_n^i)\|^2]\right)^{\frac{1}{2}}
\end{aligned}
$$

By introducing $\bar{\mu}_n$ the empirical distribution of the particles $(\bar{X}_n^i)_{i=1}^N$, the second term on the right hand side can be decomposed as the square root of the sum of two terms $A$ and $B$ defined as:

$$A = \sum_{i=1}^N \mathbb{E}[\|P_{\hat{\mu}_n}\nabla\log(\frac{\hat{\mu}_n}{\pi})(X_n^i) - P_{\bar{\mu}_n}\nabla\log(\frac{\bar{\mu}_n}{\pi})(\bar{X}_n^i)\|^2]$$

$$B = \sum_{i=1}^N \mathbb{E}[\|P_{\bar{\mu}_n}\nabla\log(\frac{\bar{\mu}_n}{\pi})(\bar{X}_n^i) - P_{\mu_n}\nabla\log(\frac{\mu_n}{\pi})(\bar{X}_n^i)\|^2]$$

By using Lemma 14, the map $(z,\mu) \mapsto P_\mu \nabla\log(\frac{\mu}{\pi})(z)$ is $L$-Lipschitz and we can bound the first term as follows :

$$A \leq \sum_{i=1}^N \mathbb{E}[\|P_{\hat{\mu}_n}\nabla\log(\frac{\hat{\mu}_n}{\pi})(X_n^i) - P_{\hat{\mu}_n}\nabla\log(\frac{\hat{\mu}_n}{\pi})(\bar{X}_n^i)\|^2] + \sum_{i=1}^N \mathbb{E}[\|P_{\hat{\mu}_n}\nabla\log(\frac{\hat{\mu}_n}{\pi})(\bar{X}_n^i) - P_{\bar{\mu}_n}\nabla\log(\frac{\bar{\mu}_n}{\pi})(\bar{X}_n^i)\|^2]$$

$$\leq \sum_{i=1}^N L^2\mathbb{E}[\|X_n^i - \bar{X}_n^i\|^2] + \sum_{i=1}^N L^2\mathbb{E}[W_2^2(\hat{\mu}_n, \bar{\mu}_n)]$$

$$= NL^2 c_n^2 + NL^2\mathbb{E}[W_2^2(\hat{\mu}_n, \bar{\mu}_n)].$$

Hence,

$$A^{\frac{1}{2}} \leq L\sqrt{N}(c_n + \mathbb{E}[W_2^2(\hat{\mu}_n, \bar{\mu}_n)]^{\frac{1}{2}}) \leq 2L\sqrt{N}c_n.$$

The second term can be bounded as:

$$B = \sum_{i=1}^{N} \mathbb{E}[\| \frac{1}{N} \sum_{i=1}^{N} (b(\bar{X}_n^j, \bar{X}_n^i) - \int b(x, \bar{X}_n^i) d\mu_n(x))\|^2]$$

$$= \sum_{i=1}^{N} \frac{1}{N^2} \sum_{j=1}^{N} \mathbb{E}[\|b(\bar{X}_n^j, \bar{X}_n^i) - \int b(x, \bar{X}_n^i) d\mu_n(x)\|^2]$$

$$\leq \sum_{i=1}^{N} \frac{1}{N^2} \sum_{j=1}^{N} L^2 \mathbb{E}[\|\bar{X}_n^j - \int x d\mu_n(x)\|^2]$$

$$\leq L^2 var(\mu_n)$$

by using Corollary 15. Hence,

$$B^{\frac{1}{2}} \leq L\sqrt{var(\mu_n)},$$

and we get the recurrence relation for $c_n$:

$$c_{n+1} \leq c_n + \frac{\gamma}{\sqrt{N}} (A + B)^{\frac{1}{2}}$$

$$\leq c_n + \frac{\gamma}{\sqrt{N}} (2L\sqrt{N} c_n + L\sqrt{var(\mu_n)})$$

$$\leq c_n(1 + 2\gamma L) + \frac{\gamma L}{\sqrt{N}} \sqrt{var(\mu_n)}$$

$$\leq \frac{1}{2} \left( \frac{1}{\sqrt{N}} \sqrt{var(\mu_0)} e^{LT} \right) (e^{2LT} - 1)$$

where the last line uses Lemma 12.

**Lemma 12.** Consider an initial distribution $\mu_0$ with finite variance. Define the sequence of probability distributions $\mu_{n+1} = (I - \gamma P_{\mu_n} \nabla \log(\frac{\mu_n}{\pi}))_{\#}\mu_n$. Under Assumption $(\mathbf{A}_1),(\mathbf{A}_2),(\mathbf{B}_1), (\mathbf{B}_2)$, the variance of $\mu_n$ satisfies for all $T > 0$ and $n \leq \frac{T}{\gamma}$ the following inequality:

$$var(\mu_n)^{\frac{1}{2}} \leq var(\mu_0)^{\frac{1}{2}} e^{TL}$$

for $L$ a constant depending on $k$ and $\pi$.

*Proof.* Denote by $x$ and $x'$ two independent samples from $\mu_n$. We have :

$$var(\mu_{n+1})^{\frac{1}{2}} = \left( \mathbb{E} \left[ \|x - \mathbb{E}[x'] - \gamma P_{\mu_n} \nabla \log(\frac{\mu_n}{\pi})(x) + \gamma \mathbb{E} \left[ P_{\mu_n} \nabla \log(\frac{\mu_n}{\pi})(x') \right] \|^2 \right] \right)^{\frac{1}{2}}$$

$$\leq var(\mu_n)^{\frac{1}{2}} + \gamma \left( \mathbb{E} \left[ \| P_{\mu_n} \nabla \log(\frac{\mu_n}{\pi})(x) - \mathbb{E} \left[ P_{\mu_n} \nabla \log(\frac{\mu_n}{\pi})(x') \right] \|^2 \right] \right)^{\frac{1}{2}}$$

$$\leq var(\mu_n)^{\frac{1}{2}} + \gamma L \mathbb{E}_{x,x' \sim \mu_n} \left[ \|x - x'\|^2 \right]^{\frac{1}{2}}$$

$$\leq var(\mu_n)^{\frac{1}{2}} + \gamma L var(\mu_n)^{\frac{1}{2}}$$

The second and last lines are obtained using a triangular inequality while the third line uses that $x \mapsto P_{\mu_n} \nabla \log(\frac{\mu_n}{\pi})(x)$ is $L$-Lipschitz by Lemma 14. We then conclude using Lemma 13. $\square$

**Lemma 13.** [Discrete Gronwall lemma] Let $a_{n+1} \leq (1 + \gamma A)a_n + b$ with $\gamma > 0$, $A > 0$, $b > 0$ and $a_0 = 0$, then:

$$a_n \leq \frac{b}{\gamma A} (e^{n\gamma A} - 1).$$

*Proof.* Using the recursion, it is easy to see that for any $n > 0$:

$$a_n \leq (1 + \gamma A)^n a_0 + b \left( \sum_{i=0}^{n-1} (1 + \gamma A)^k \right)$$

One concludes using the identity $\sum_{i=0}^{n-1} (1 + \gamma A)^k = \frac{1}{\gamma A}((1 + \gamma A)^n - 1)$ and recalling that $(1 + \gamma A)^n \leq e^{n\gamma A}$. $\square$

## 12   Auxiliary results

**Lemma 14.** Under Assumption $(\mathbf{A}_1)$,$(\mathbf{A}_2)$,$(\mathbf{B}_1)$, $(\mathbf{B}_2)$, the map $(z,\mu) \mapsto P_\mu \nabla \log(\frac{\mu}{\pi})(z)$ is $L$-Lipschitz with:

$$\|P_\mu \nabla \log(\frac{\mu}{\pi})(z) - P_{\mu'} \nabla \log(\frac{\mu'}{\pi})(z')\| \le L(\|z - z'\| + W_2(\mu,\mu')) \tag{40}$$

where $L$ depends on $k$ and $\pi$.

*Proof.* We will consider an optimal coupling $s$ with marginals $\mu$ and $\mu'$:

$$
\begin{aligned}
\|P_\mu \nabla \log(\frac{\mu}{\pi})(z) - P_{\mu'} \nabla \log(\frac{\mu'}{\pi})(z')\| &= \| \mathbb{E}_s \left[ \nabla \log \pi(x) k(x,z) - \nabla \log \pi(x') k(x',z') \right] \\
&\quad + \mathbb{E}_s \left[ \nabla_1 k(x,z) - \nabla_1 k(x',z') \right] \| \\
&\le B \mathbb{E}_s \left[ \| \nabla \log \pi(x) - \nabla \log \pi(x') \| \right] + C_V \mathbb{E}_s \left[ \| k(x,z) - k(x',z') \| \right] + \mathbb{E}_s \left[ \| \nabla_1 k(x,z) - \nabla_1 k(x',z') \| \right] \\
&\le BM \mathbb{E}_s [\|x - x'\|] + C_V D \left( \|z - z'\| + \mathbb{E}_s[\|x - x'\|] \right) + D \left( \|z - z'\| + \mathbb{E}_s[\|x - x'\|] \right) \\
&\le L(\|z - z'\| + W_2(\mu,\mu'))
\end{aligned}
$$

The second line is obtained by convexity while the third one uses Assumption $(\mathbf{B}_1)$ and $(\mathbf{A}_1)$. The penultimate one uses Assumption $(\mathbf{A}_2)$ and $(\mathbf{B}_2)$; finally the last line relies on $s$ being optimal and setting $L = C_V(D+1) + BM$. □

**Corollary 15.** Let $b$ the function defined by $b(x,z) = \nabla \log \pi(x) k(x,z) + \nabla k(x,z)$. Under the assumptions of Lemma 14, $b$ is $L$-Lipschitz in its first variable.

*Proof.* Notice that $P_\mu \nabla \log(\frac{\mu}{\pi})(y) = \mathbb{E}_{x \sim \mu}[b(x,z)]$ for any $\mu \in \mathcal{P}_2(\mathcal{X})$ and $z \in \mathcal{X}$. Hence, for any $y, y' \in \mathcal{X}$,

$$|b(y,.) - b(y',.)| \le L W_2(\delta_y, \delta_{y'}) = L\|y - y'\|. \qquad \square$$

**Lemma 16.** Suppose Assumption $(\mathbf{A}_1)$ holds, i.e. the kernel and its gradient are bounded by some positive constant $B$. Moreover, assume that $\nabla \log(\pi)$ is $M$-Lipschitz and that $\int \|x\| \mu_n(x) dx$ is uniformly bounded on $n$. Then $I_{Stein}(\mu_n|\pi)$ remains bounded by some $C > 0$, i.e. Assumption $(\mathbf{A}_3)$ holds.

*Proof.* For any $\mu$, we have :

$$I_{Stein}(\mu|\pi) = \langle \int \nabla \log \pi(x) k(x,.) + \nabla_1 k(x,.) d\mu(x), \int \nabla \log \pi(y) k(y,.) + \nabla_1 k(y,.) d\mu(y) \rangle_{\mathcal{H}}$$

Using the reproducing property and integration by parts it is possible to write $I_{Stein}(\mu|\pi)$ as:

$$
\begin{aligned}
I_{Stein}(\mu|\pi) = &\int \nabla_1 . \nabla_2 k(x,y) d\mu(x) d\mu(y) \\
&+ \int \langle \nabla \log \pi(y), \nabla_1 k(x,y) \rangle d\mu(x) d\mu(y) + \int \langle \nabla \log \pi(x), \nabla_1 k(y,x) \rangle d\mu(x) d\mu(y) \\
&+ \int \langle \nabla \log \pi(x), \nabla \log \pi(y) \rangle k(x,y) d\mu(x) d\mu(y).
\end{aligned}
$$

The terms involving the kernel are easily bounded since the kernel is bounded with bounded derivatives. Using that $\nabla \log \pi$ is $M$-Lipschitz, it is easy to see that

$$\|\nabla \log \pi(x)\| \le \|\nabla \log \pi(0)\| + M\|x\|. \tag{41}$$

Using the above inequality, one can directly conclude that $\int \|x\| \mu_n(x) dx$ remains bounded. □

## 13   Experiments

We downloaded and reused the code (in Python) from Liu and Wang (2016) available at `https://github.com/dilinwang820/Stein-Variational-Gradient-Descent` for our experiments. It implements a toy example with a 1-D Gaussian mixture and a gaussian kernel. In the upper figures, the blue dashed lines are the target density function and the solid green lines are the densities of the (200) particles at different iterations of our algorithm (estimated using kernel density estimator). The lower figures represent the evolution of $I_{Stein}(\hat{\mu}_n|\pi)$ and $\hat{KL}(\hat{\mu}_n|\pi)$[3] along iterations $n \ge 0$. One can see on the upper figures that the particles recover the target distribution. On the lower figure (in log-log scale) , one can see that the average $I_{Stein}$ over $n$ iterations (i.e. $1/n \sum_{k=1}^n I_{Stein}(\hat{\mu}_k|\pi)$) decreases at rate $1/n$ as predicted in Corollary 6. The code to reproduce our results is available : `https://github.com/akorba/SVGD_Non_Asymptotic`.

Figure 1: The particle implementation of the SVGD algorithm illustrates the convergence of $I_{Stein}(\mu_n|\pi)$ to 0.