[Reviews · NeurIPS 2020]

Review 1

Summary and Contributions: The authors consider Stein Variational Gradient Descent (SVGD) algorithm, which optimises a set of particles to approximate a target probability distribution. The authors under slightly weaker assumption establish that algorithm is decreasing. And in next shows convergence of infinite particle system under Stein log-Sobolev inequality assumption.

Strengths: The main result is the convergence of infinite SVGD with infinite particle to target, which reduces gap between theory and practical implementation.

Weaknesses: The result about finite (in appendix) particle depends exponentially on parameters which make results non-applicable in practice.

Correctness: The proofs seem to be correct.

Clarity: The paper is quite well written.

Relation to Prior Work: The authors nicely present existing results in the field and clearly explain their contribution.

Reproducibility: Yes

Additional Feedback: My detailed minor remarks: - In chapter 2: $$L_2(\mu)$$ should be a space of functions $$f : \mathcal{X} \rightarrow \mathbb{R}$$, not $$f: \mathcal{X} \rightarrow \mathcal{X}$$. - In chapter 2: It would be nice to include some nice reference about W_2 Hessian of KL - I am not familiar with it. - In chapter 3: Praised be the author who uses the term "positive semi-definite kernel" instead of "positive definite kernel". - In section 3.2 there is inconsistency with notation once we have $$P_{\mu_n} \nabla$$, otherwise we have $$P_{\mu_n} \nabla_{W_2}$$. #UPDATE After reading rebuttal and other reviews I still reccomend to accept.


Review 2

Summary and Contributions: This paper proves linear convergence in KL of the discretized, infinite-particle version of Stein variational gradient descent (SVGD) under a Stein log Sobolev inequality (plus additional technical assumptions).

Strengths: The paper proves an important -- although not too surprising result -- about the convergence of infinite-particle SVGD. Overall, the analysis is elegant and other than the Stein LSI, only relies on very reasonable smoothness assumptions, plus a single boundedness assumption. The SM (Prop 12) also provides an important propagation of chaos result on the finite-particle accuracy of SVGD. This is a valuable paper for understanding the approximation properties of SVGD and similar algorithms.

Weaknesses: There needs to be more discussion of examples where all three assumptions required for Prop 5 hold. One concern I have is that these assumptions (particularly A1) are in conflict with the Stein LSI holding. Do A1 hold in the 1D examples from Duncan et al. 2019 discussed after Definition 2? Is there any hope of checking whether A3 holds?

Correctness: While I have not checked the proofs in detail, the claims appear to be correct and reasonable.

Clarity: Yes

Relation to Prior Work: Yes

Reproducibility: Yes

Additional Feedback: ********** Update based on author rebuttal: The authors did a good job addressing my comments. I'm happy to accept this paper and increased my score to 7. Though I very much hope the authors move their finite-particle results to the main paper. ********** A. I would encourage the authors to add at a least a short presentation of the finite-particle results to the main paper -- these are important. B. Please be sure to put citations in parentheses when appropriate. There were many confusing non-parenthetical citations at the ends of sentences.


Review 3

Summary and Contributions: This paper proves the first nonasymptotic convergence bound for the SVGD, hence provides a nonasymptotic analysis of the SVGD (for its infinite particle limit) for the first time in the literature.

Strengths: The paper is well written, the bounds are clean and understandable. SVGD has become a popular method, hence its analysis is relevant for the NeurIPS community.

Weaknesses: As for most of the analysis papers, I do not think verifying whether the target satisfies the log-Sobolev inequality is not trivial, hence the bounds in this paper may not be immediately practical yet. Furthermore, the analysis applies to the infinite-particle limit and finite-bounds are not as clean as the infinite-particle case. (see my detailed comments)

Correctness: Yes.

Clarity: Yes. In many places, citations are not formatted correctly, e.g., they are always of the form Author (year), whereas in many places where it should be (Author, year).

Relation to Prior Work: Yes. Previous contributions in this area focused on either a continuous-time analysis or asymptotic convergence results. This paper provides a first nonasymptotic result.

Reproducibility: Yes

Additional Feedback: Post rebuttal: Thank you for you response. I keep my score as it is (which is accept) -- but like R2, I really would like to see the finite-N results in the extra page that would be provided if this paper is accepted. -- This paper deals with an important open problem so far, deriving the convergence rates of the SVGD in discrete-time. The analysis looks sound and nicely done. (1) First of all, the key to establish the results is the assumption that the target satisfies Stein log-Sobolev inequality. Some elaboration on what satisfies this inequality is provided before Sec. 3.4, but it would be especially nice if somewhat a realistic example is given for which the theory provably holds. (2) The convergence rate is obtained in the number of iterations in discrete-time, but in the infinite-particle limit. Therefore, these results are still not precisely about the SVGD, but on its infinite-particle limit. Results regarding the finite-particle case are deferred to Appendix but I am still not sure what they mean. Is it possible to obtain a unified convergence bound, containing N as in Prop. 12 (the number of particles) and n (the number of iterations) as in Theorem 7 together? Currently these two results are bounding different things and it is not clear a finite-time meaningful result (which is decaying in n *and* N) can be obtained either in KL or in Wasserstein-2 distance. This is especially the case since the bound in Prop. 12 is time-growing as opposed to Theorem 7, which is geometrically decaying in the number of iterations. So it looks like time-decaying nature of Theorem 7 does not apply to the finite N case, which should be made explicit as an open problem. Without this final result in the paper, I actually think the title should be slightly edited as well, as the analysis applies only to the infinite-particle limit case, rather than the real SVGD algorithm with finite number of particles. (3) A discussion related to the effect of dimensions in the bounds is missing. In particular, I strongly suggest authors to identify the dimension dependence (qualitatively, if not quantitatively) in bounds and comment on whether these bounds deteriorate as d grows, if so, how fast. Again, relating to comment (2), it is certain that the finite-particle bound would suffer from dimensionality but it would be interesting to also see whether the infinite-particle bound has any bad dependence to d. (4) I am not sure about the sentence following Theorem 7: "Hence, if 2cγ λ < 1, we obtain exponential convergence." It looks like it is not possible to convert this sentence into a condition on the step-size and *guarantee* that this holds. So, to be clear, is it possible that despite the conditions in Theorem 7 being satisfied, the above quantity can be bigger than 1? This would result in a bound growing in time. If this is the case, I also strongly suggest some discussion on this as to how/when this can happen.


Review 4

Summary and Contributions: This paper study the Stein Variational Gradient Descent (SVGD) algorithm, a particle variational inference method, which optimizes a set of particles to approximate a target probability distribution. This algorithm has shown to be practical but the non-asymptotic analysis remains incomplete. The most interesting part of this paper is it provides a guarantee on the convergence rate in KL divergence, under some minor assumptions. They argue that, with recent results on the Stein geometry, they can draw a clear connection between time-discretized Wasserstein gradient flow and SVGD. Compared to recent related works, more natural and simple assumption is needed and the analysis looks simple yet correct.

Strengths: [+] detailed discussions about the related works and background. [+] clear discussion and summarization about the theoretical result. [+] Well-written.

Weaknesses: [-] In section 5, the authors provide an analysis of the discretization of the gradient flow. The assumption A1 - A3 looks not so intuitive, i hope some remarks e.g. connection to moment constraints, can be discussed here. [-] Line 312 -313 says theorem 7 'gives clearer connections with Wasserstein gradient flows'. Here, a revisit of the boundedness condition of KSD may make the comparsion clearer.

Correctness: the claims and methods are correct.

Clarity: This paper is well-written.

Relation to Prior Work: This paper clearly discusses how this work differs from previous contributions.

Reproducibility: Yes

Additional Feedback:

[Author Response · NeurIPS 2020]

We thank **R1, R2, R3, R4**, who provided overall scores (7-6-7-7) respectively, for their careful reading of the paper,
their positive comments on its clarity and interest, and their relevant questions. We recall that our paper provides rates
of convergence for the iterates $(\mu_n)_{n \geq 0}$ of SVGD in the infinite particle regime (Eq 14) (i.e., of the time-discretized
version of the SVGD gradient flow $\mu_t$ (Eq 11)) to a target distribution $\pi \propto \exp(-V)$. At time $n$, the distribution
$\mu_n$ is associated to a Mac-Kean Vlasov process $x_n$ whose dynamics depends on $\mu_n$ itself. Therefore, the practical
implementation of SVGD relies on approximating $x_n$ with $N$ interacting particles $(\hat{x}_n^i)_{i=1}^N$ (Eq 38), where the empirical
distribution $\hat{\mu}_n$ of the particles approximates $\mu_n$. In particular, we provide a $\mathcal{O}(1/n)$ convergence rate for the arithmetic
mean of the Kernel Stein Discrepancy (KSD) (which metricizes weak convergence in many cases, see Sec. 3.3) between
the iterates $\mu_n$ and $\pi$, under Assumptions A1–A3. It **does not rely on Stein LSI nor on convexity of** $V$, unlike most
of the results on Langevin Monte Carlo (LMC) which assume either (standard) LSI or convexity of $V$.

**R1, R3. Finite number of particles.** We would like to clarify our result (Prop 12). It is non-asymptotic one in the
sense that it provides an explicit bound. However, *it is not a bound that helps quantify the rate of minimization of the*
*objective function*. Rather, it is a bound between the population distribution $\mu_n$ and its particle approximation $\hat{\mu}_n$. It
states that for a fixed time horizon $T > 0$, $\mathbb{E}[W_2^2(\mu_n, \hat{\mu}_n)] \leq C\frac{1}{\sqrt{N}}$, where $N$ is the number of particles. Such results
are referred to *propagation of chaos* in the PDE literature, where having $C$ depending on $T$ is common. Getting a
similar bound with $C$ *not* depending on $T$ would be a much stronger result referred to as *uniform in time propagation of*
*chaos* (see l.522-525). Such results, which are subject to active research in PDE, are hard to obtain. Among the recent
exceptions is (Durmus, 2018a) who consider the process $dx_t = -\nabla U(x_t) - \nabla W * \mu_t(x_t)dt$ and manage to prove such
results when $U$ is strictly convex outside of a ball. However in SVGD (see Eq 8), the attractive force $\nabla \log \pi(x)k(x, .)$
cannot be written as the gradient of a confinement potential $U$ in general. Hence these results do not apply. **R3 (2).**
In our answer to R2 below, we discuss a contradiction in the Th 7 assumptions, discovered by R2. We will therefore
acknowledge that the convergence rate for SVGD using $\hat{\mu}_n$ remains an open problem.

**R2, R4. Assumptions A1-A3.** A1 and A2 are mild smoothness assumptions on $(k, V)$. In particular, A2 is standard in
the LMC algorithm literature. A1 is needed to obtain our core descent result, Prop 5, because KL is not smooth, see
Remark 3 l.320-323. A3 can be checked in each specific context. For instance, in Lemma 17, we provide conditions
under which A3 holds. The validity of this hypothesis is also confirmed in our experiments.

**R2.** Thank you for raising the question of whether there exists a distribution $\pi$ and kernel $k$ that simultaneously
satisfy the Stein LSI and assumption A1. Having thought about this, *we believe that no such $\pi$ and $k$ exist* (at
least for $\mathcal{X} = \mathbb{R}^d$). Given that both the kernel and its derivative are bounded, equation $\int \sum_{i=1}^d [(\partial_i V(x))^2 k(x, x) -$
$\partial_i V(x)(\partial_i^1 k(x, x) + \partial_i^2 k(x, x)) + \partial_i^1 \partial_i^2 k(x, x)]d\pi(x) < \infty$ reduces to a property on $V$ which, as far as we can tell,
always holds; and this implies that Stein LSI does not hold (see l.163-165). For instance, even when $V = -\log(\text{cauchy})$
or $V = -\log(\text{student})$, we quickly find that the resulting expectations are bounded. We will therefore remove Th 7
from our results, and replace it with a discussion on the difficulty in simultaneously ensuring conditions for a descent
lemma and for Stein LSI. In particular, we recall that in the classical LMC setting, we would require only smoothness
assumptions on $V$ (A2) and the classic LSI inequality to obtain exponential convergence in the objective (here KL), see
(Vempala, 2019). This is also the approach in nonconvex optimization (where LSI is called Polyak-Lojasiewicz (PL)
inequality, see Rk 3). Unfortunately, as KL is not smooth (see l.320-323), we had to further assume A1 i.e. boundedness
of the kernel in Prop. 5, resulting in the contradiction in Thm. 7. We emphasize that Corollary 6 establishes convergence
under very general conditions, and remains valid, however we cannot now show fast rates. **R2, R3. Validity of the**
**Stein LSI itself.** (Duncan et al., 2019) discusses conditions for which the Stein LSI on its own is satisfied, among
which are the 1-d examples provided Sec 3.3 (which, unfortunately, do not satisfy A.1). Construction of examples
satisfying Stein LSI should begin by ensuring that l.164 does not hold, i.e. the integral is infinite (see above): eg, $k$
polynomial of order 3 or greater and $\pi$ with exploding $\beta$-moments, for $\beta \geq 3$ (e.g., a student distribution in $\mathcal{P}_2(\mathcal{X})$).

**R1.** The Wasserstein Hessian of $\text{KL}(.|\pi)$ is briefly mentioned in (Villani, 2003, Sec 8.2) and (Wibisono, 2018, Sec
3.1.1) but was not particularly highlighted in the ML literature. We derive all the formulas in the proof of Prop 5.

**R3. (3)** We will correct this with a precise discussion about the dependence on the dimension $d$. In Cor. 6 and
Th. 7, the constants $M, B, C, \text{KL}(\mu_0|\pi)$ are parameters of the problem and depend indeed implicitly on $d$. To
explicit the dependence of $\text{KL}(\mu_0|\pi)$ on $d$, we can apply (Vempala, 2019, Lem. 1): under A2, we have $\text{KL}(\mu_0|\pi) \leq$
$V(x_\star) + \frac{d}{2} \log\left(\frac{M}{2\pi}\right)$, where $x_\star$ is a stationary point of $V$ and assuming that $\mu_0 \sim \mathcal{N}(x_\star, \frac{1}{M})$. We will explicit
the dependence e.g. for $M, B$ for a gaussian kernel $k$ and quadratic potential $V$ for illustration. **(4)** Assuming
$0 < \gamma < \min\left(\frac{1}{2\lambda}, \frac{1}{B^2(\alpha^2+M)}\right)$ is sufficient to obtain $0 < 2c_\gamma \lambda < 1$. Indeed, $\gamma < \frac{1}{B^2(\alpha^2+M)}$ implies $\frac{\gamma}{2} < c_\gamma$. Since
$c_\gamma < \gamma$, we have $0 < \lambda\gamma < 2c_\gamma\lambda < 2\gamma\lambda < 1$, using the assumption on $\gamma$.

**R4.** SVGD can be seen as a gradient descent (GD) algorithm to minimize $\text{KL}(\cdot|\pi)$. In this context, the KSD
($I_{stein}(\mu_n|\pi)$) plays the role of the squared norm of the gradient at time $n \geq 0$. Assumption A3 is analogous to
assuming $\sup_n \|\nabla F(x_n)\|^2 < \infty$ when applying GD to minimize some function $F$ over $\mathbb{R}^d$, which is a standard
bounded gradient assumption in optimization. It holds for instance if the objective function $F$ is Lipschitz.

[Meta-Review · NeurIPS 2020]

The reviewers agree that this paper would make a valuable contribution to NeurIPS. Please have a look at the reviewer comments (especially about moving finite-particle results to the main paper) while working on the final manuscript.